# Insulin-like Peptides of the Western Flower Thrips *Frankliniella occidentalis* and Their Mediation of Immature Development

**DOI:** 10.3390/insects14010047

**Published:** 2023-01-03

**Authors:** Chul-Young Kim, Yong-Gyun Kim

**Affiliations:** Department of Plant Medicals, College of Life Sciences, Andong National University, Andong 36729, Republic of Korea

**Keywords:** *Frankliniella occidentalis*, insulin-like peptide, growth, insulin signaling, hot pepper

## Abstract

**Simple Summary:**

The western flower thrips, *Frankliniella occidentalis,* is an invasive and polyphagous insect pest. Its regional and host diversity suggests variation in its developmental rate. In addition, a previous study indicated that insulin signaling is related to thrips development. However, no insulin-like peptide (*ILP*) was identified in *F. occidentalis*. This study identified two *ILP* genes (*Fo-ILP1* and *Fo-ILP2*) and their expression profiles were investigated. Further, their expressions were dependent on feeding activity, and the starvation treatment greatly reduced the expression levels of both *ILP* genes. The RNA interference of the *ILP* gene expressions had adverse effects on the immature development of the thrips. However, different host plants causing variation in developmental rates also led to variation in the *ILP* gene expression levels. These results report the first *ILP* genes in thysanopteran insects and their functional relationship with the thrips’ development.

**Abstract:**

Insulin-like peptides (*ILP*s) mediate various physiological processes in insects. Specifically, *ILP* expression is required for immature development in different insects. The western flower thrips, *Frankliniella occidentalis,* is polyphagous, but its occurrence and population density vary among different hosts. This study assesses the developmental variations in the thrips through quantitative analysis of their *ILP* expressions. Two types of *ILP*s (*Fo-ILP1* and *Fo-ILP2*) were identified from the genome of *F. occidentalis*, and both *ILP*s were predicted to have the characteristics of signal peptides and B-C-A chains linked by cysteines. A phylogenetic analysis indicates that these two *ILP*s in the thrips are clustered with the *ILP1* of *Drosophila melanogaster*, suggesting their physiological roles in growth. In addition, the two *ILP* genes were relatively highly expressed at all feeding stages, but their expression was reduced during the nonfeeding prepupal and pupal stages. Furthermore, RNA interference of each *ILP* expression led to significant developmental retardation. In validating the *ILP* expression in the thrips’ development, five different varieties of host hot peppers were assessed in a choice test, along with the immature development of *F. occidentalis*. The expression levels of the two *ILP* genes were highly correlated with variations in the immature developmental rates of different hot pepper varieties. These suggest that *Fo-ILP1* and *Fo-ILP2* mediate the immature development of *F. occidentalis* by sensing different nutritional values of hot peppers. This study is the first report on *ILP*s in thysanopteran insects.

## 1. Introduction

Insulin signaling is required for insect development, similar to other metazoans. Insulin-like peptides (*ILP*s) and their signaling mechanisms have been extensively assessed in various insects. The first insect *ILP* was discovered unexpectedly during the identification of the prothoracicotropic hormone (PTTH) of *Bombyx mori,* known as bombyxin [1]. Later, 38 bombyxins were identified from its genome analysis [2]. *Drosophila melanogaster* encodes seven *ILP* genes [3]. Thus, most insects are likely to have multiple *ILP* genes [4]. Similar to mammalian insulin, insect *ILP*s consisting of three chains undergo post-translational modifications by proteolytic cleavage to produce mature heterodimeric peptides consisting of A-B chains linked by disulfide bonds [5].

Insulin signaling controls the blood sugar level, growth, and reproduction in vertebrates [6]. However, the insulin-mediated physiological processes are triggered by insulin synthesized in the β-cells of the pancreas, insulin-like growth factors synthesized by the liver in response to stimulatory signals of the growth hormone, or relaxin produced by the reproductive organs. The insect *ILP*s are produced in the brain, specifically in the insulin-producing cells (IPCs) in the pars intercerebralis of the brain. After food ingestion, the fat body can sense nutrients and send a nutritional signal called the “fat-body-derived signal” [7] to the IPCs in *Drosophila*. The secreted *ILP*s reach the target cells and trigger the insulin signaling pathway by binding to a membrane receptor called the insulin receptor (*InR*).

The western flower thrips *Frankliniella occidentalis* feeds on a large number of host plants and transmits a serious plant pathogen, the tomato spotted wilt virus (TSWV) [8]. Although the thrips feed on different hosts, they show host preference and subsequent developmental variations with different hosts [9,10,11]. Based on insulin signaling in thrips, an insulin receptor, *Fo-InR*, was identified in a previous study [12]. This study showed that *Fo-InR* encodes a sequence of 1645 amino acids, which is highly homologous (>70% homologies) to known insect *InR*s [12]. *Fo-InR* was expressed in most developmental stages, while starvation upregulated the expression levels. Furthermore, expression analysis in thrips reared on different diets indicates that the *InR* expression levels are negatively correlated with the immature developmental rate of *F. occidentalis*. This suggests that insulin signaling mediates the growth and development of *F. occidentalis*. However, no *ILP*s have been identified in thysanopteran insects, including *F. occidentalis*.

This study predicted two *ILP* genes in the *F. occidentalis* genome and analyzed the expression patterns to confirm their association with the development of the thrips. Additionally, using RNA interference, the physiological functions during immature growth were assessed. Furthermore, the expressional variations of the two *ILP* genes for different host plant varieties were explained in accordance with the host preference and subsequent developmental rates.

## 2. Materials and Methods

### 2.1. Insect Rearing and Host Plant Culture

A laboratory population of *F. occidentalis* was obtained from the Department of Crop Protection, National Institute of Agricultural Sciences (Jeonju, Republic of Korea), and cultured under laboratory conditions of 25 ± 2 °C, 16:8 h (L:D), and 60 ± 5% relative humidity. The germinated beans (*Phaseolus coccineus* L.) were supplied for feeding and oviposition. The newly laid eggs on the beans in adult colonies were transferred to a breeding dish (SPL Life Sciences, Pochon, Republic of Korea). After hatching, beans were supplied every day to the breeding dish. Under the laboratory conditions, the larvae underwent two instars (L1–L2) that were distinct from prepupae or pupae with wing pads.

The hot pepper (*Capsicum annuum*) seeds were purchased from different companies (Appendix A) and germinated in a nursery plate. The resulting seedlings after 2 weeks were transferred to pots (5 cm in diameter and 8 cm in height) and maintained under the thrips rearing conditions. The bioassay leaves for feeding were obtained from hot pepper plants with a height of 30 cm.

### 2.2. Bioinformatics to Predict ILP Genes in F. occidentalis

The amino acid and DNA sequences of two *ILP* genes (*Fo-ILP1* and *Fo-ILP2*) of *F. occidentalis* were obtained from the National Center for Biotechnology Information (NCBI; https://blast.ncbi.nlm.nih.gov) (accessed on 31 December 2022) with accession numbers as shown in Appendix A. MEGA6.0 was used for the phylogenetic tree by maximum likelihood, where the evolutionary distances were computed using the Poisson correction method. Bootstrapped values were obtained with 1000 replications to support branching and clustering. The N-terminal signal peptides of *Fo-ILP1* and *Fo-ILP2* were determined using the SignalP 5.0 server (https://services.healthtech.dtu.dk/service.SignalP-5.0/) (accessed on 31 December 2022). Subsequently, three different chains of *ILP*s were predicted from the *ILP* templates of *D. melanogaster* [13]. These chains were confirmed by prediction of the putative cleavage sites using a program developed by the Neuroproteomics and Neurometabolomics Center on Cell–Cell Signaling (http://stagbeetle.animal.uiuc.edu/cgi-bin/neuropred.py) (accessed on 31 December 2022).

### 2.3. RNA Extraction and cDNA Preparation

The RNAs of *F. occidentalis* were extracted from different developmental stages using Trizol reagent (Invitrogen, Carlsbad, CA, USA) according to the manufacturer’s instructions. Each extracted RNA was then resuspended in 30 μL of diethyl pyrocarbonate and quantified using a spectrophotometer (Nanodrop, Thermo Fisher Scientific, Wilmington, DE, USA). In addition, about 400 ng of the RNA were used for cDNA synthesis using an RT premix (Intron Biotechnology, Seoul, Republic of Korea) containing the oligo dT primer according to the manufacturer’s instructions.

### 2.4. RT-PCR and RT-qPCR

The prepared cDNAs were used for RT-PCR with Taq polymerase (GeneALL, Seoul, Republic of Korea) and gene-specific primers (Appendix A). After initial denaturation at 95 °C for 5 min, PCR was performed using a thermocycler (Step One Plus Real-Time PCR System; Applied Biosystems) with 35 amplification cycles (95 °C for 1 min, 50–55 °C for 30 s, and 72 °C for 1 min) and finalized with an additional extension step at 72 °C for 10 min. The PCR products were assessed by 1% agarose gel electrophoresis. RT-qPCR was performed as per the procedures noted by Bustin et al. [14] using Power SYBR Green PCR Master Mix (Toyobo, Osaka, Japan) with gene-specific primers (Appendix A). In addition, after initial heat treatment at 95 °C for 2 min, qPCR was performed with 40 cycles of denaturation at 95 °C for 30 s, annealing at 50−55 °C for 30 s, and extension at 72 °C for 30 s. The elongation factor 1 (*EF1*, Appendix A) was used as a reference gene to normalize the expression level of each qPCR sample. Further, quantitative analyses were performed using the comparative CT (2^−ΔΔCT^) method [15]. All experiments were independently replicated three times.

### 2.5. RNA Interference (RNAi) of ILP Genes Using Gene-Specific Double-Stranded RNA (dsRNA)

The template DNAs of the two *ILP* genes were amplified into 316 base pairs (bp) for *Fo-ILP1* and 201 bp for *Fo-ILP2* using cDNA with gene-specific primers (Appendix A) containing the T7 promoter sequence (5′-TAATACGACTCACTATAGGGAGA-3′) at the 5′ end. The PCR products were used for the in vitro synthesis of the dsRNAs with the NTP mixture at 37 °C for 3 h using T7 RNA polymerase in the MEGAscript RNAi kit (Ambion, Austin, TX, USA). The resulting dsRNAs were mixed with a transfection reagent, Metafectene PRO (Biontex, Plannegg, Germany), at a 1:1 (*v*/*v*) ratio, and incubated at 25 °C for 30 min to form liposomes that protect the dsRNA from attack by the digestive enzymes. This dsRNA suspension (200 ng/µL) was used to evenly cover a bean diet with 10 μL volume, to which 25 test thrips at the L1 larval stage were applied and allowed to feed for 12 h. Thereafter, the fresh and untreated diets were replaced. A control dsRNA (dsCON) specific to enhanced green fluorescence protein (*EGFP*) was prepared according to the method of Vatanparast and Kim [16].

### 2.6. Screening Hot Pepper Varieties by Assessment of Feeding Damage

Five different hot pepper varieties (S1, S2, and R1–R3) (Appendix A) were cultured in pots until the four-leaf stage. These were commercially procured and had different susceptibilities to TSWV; S1 and S2 were susceptible, whereas R1–R3 were resistant to TSWV infection. A total of 50 hot peppers, with 10 plants per variety, were contained in 100 L-size screen cages (Gaonunri, Seoul, Republic of Korea). Hundred adults of *F. occidentalis* were released into these cages and allowed to feed on the hot peppers for 1 week under the rearing conditions described above. The damage intensity was evaluated on a scale of 6 grades (Appendix A): no damage for grade 0, small damage limited to one leaf for grade 1, two leaves damaged on the upper 1/3 for grade 2, two leaves damaged on the upper 1/2 for grade 3, more than 2 leaves damaged for grade 4, and all leaves damaged with a yellowish color for grade 5.

### 2.7. Choice Test Using a Two-Way Tunnel Assay

A two-way tunnel assay was used to determine the choice test for flying adults [17]. Using this method, two different hot pepper hosts were compared to determine the choices of the *F. occidentalis* adults. The arena consisted of two chambers of identical sizes (25 × 25 × 25 cm) for setting two hot pepper plant varieties for the choice test (illustrated in Figure 4B). The distance between the two chambers was 50 cm. In each test, 100 adults were released through an introduction hole to observe their choices. The yellow sticky traps (15 × 10 cm, Green-AgroTech, Gyeongsan, Republic of Korea) were installed above the test hot peppers to collect the thrips. The tests were performed for 24 h at 25 ± 2 °C and 70 ± 5% relative humidity under 1000 lux illumination.

### 2.8. Assessment of Immature Development Period and Survival Rate for Two Hot Pepper Varieties

The thrips were reared on two different hot pepper varieties (R1 and R2) with a control diet for rearing thrips from L1 to adult under the rearing conditions described above. Newly hatched L1 larvae were placed on a disc (4 cm in diameter) of hot pepper leaves or a control diet in the insect breeding dish. In order to prevent desiccation, a wet filter paper was placed in the dish. The developmental stages or survival were observed under a microscope (EZ4, Leica, Wetzlar, Germany) every day at 10 p.m. Each treatment used 20 thrips with three replications. The developmental rate was estimated as the inverse of the developmental period from L1 to adult emergence.

### 2.9. Correlation Analysis between Developmental Rate and ILP Expression

The developmental rates under three different diet conditions (R1, R2, and control diet) were compared with the expression levels of the two *ILP* genes (*Fo-ILP1* and *Fo-ILP2*). The gene expression levels were estimated by RT-qPCR, as described above. However, for the RT-qPCR, the RNAs were extracted on the first days of the L2 larvae, pupae, and adults. Each RNA sample used 25 individuals, and each treatment was replicated three times. Additionally, in the correlation analysis, the expression levels at the pupal stage were compared with the developmental rates.

### 2.10. Statistical Analysis

The percentage data were arcsine-transformed to be normalized, and data obtained from the feeding damage or tunnel tests were subjected to a one-way analysis of variance using PROC GLM in SAS [18]. The means were compared using the least significant difference (LSD) test with a type I error of 0.05. The frequency data were analyzed by the mean difference test using PROC FREQ at a type I error of 0.05 (*) and 0.01 (**).

## 3. Results

### 3.1. ILPs of F. occidentalis (Fo-ILPs) and Their Expression Profiles

The two *ILP* genes (*Fo-ILP1* and *Fo-ILP2*) were obtained from the genome of *F. occidentalis* (Figure 1A). The phylogenetic analysis indicated that these two *ILP*s are different from lepidopteran *ILP*s and closely related to those of *D. melanogaster* (*Dm-ILP*s), in which they were clustered with *Dm-ILP1*. In comparison to *Dm-ILP*s, the functional domains of the two thrips *ILP*s were predicted (Figure 1B). The cysteines and cleavage sites were conserved among the *ILP*s, which allowed the *ILP*-characteristic three chains (B-C-A) and signal peptides to be predicted.

The two *ILP* genes were expressed at all developmental stages, from larvae to adults (Figure 2A). With the exception of the prepupal and pupal stages, the *ILP* genes were expressed at relatively stable levels at the larval and adult stages. In investigating the low level of *ILP* expression during the nonfeeding stages, the larvae (Figure 2B) and adults (Figure 2C) were subjected to starvation. The starvation treatment significantly (*p* < 0.05) suppressed the gene expressions of the *ILP* genes.

### 3.2. RNAi of ILP Gene Expression Led to Developmental Retardation

In the feeding process, with gene-specific dsRNA, the expression levels of *Fo-ILP1* and *Fo-ILP2* (Figure 3A) were suppressed. Additionally, in both genes, significant suppression began to occur 12 h after RNAi treatment, and the suppressed levels were maintained for at least another 12 h. Under this RNAi condition, the immature period (=larva + pupal period) was estimated (Figure 3B). Further, both RNAi treatments led to significant (*p* < 0.05) delays in immature development in adults.

### 3.3. Developmental Variations of F. occidentalis for Different Varieties of Hot Pepper

Five different varieties of hot peppers were assessed for the thrips’ preference and development. These varieties were different in terms of TSWV infection susceptibility, with two susceptible (S1 and S2) and three resistant (R1–R3) varieties. To estimate the feeding preferences, the thrips adults were released into the five varieties and assessed for feeding damage (Figure 4A). However, a TSWV-resistant strain (R2) was highly susceptible to damage from thrips feeding, while a TSWV-susceptible strain (S1) was highly resistant to feeding damage. The preferences were further supported by the choice test (Figure 4B). The two varieties (S2 and R2) that were susceptible to feeding damage were preferentially favored by the thrips.

The two TSWV-resistant strains (R1 and R2) were compared in the assessment of thrips development (Figure 5). In this assay, we used a soybean-based diet as the control. Compared to control or R2 (favored by thrips in the above assays), thrips feeding on R1 (less favored by thrips) suffered from high immature mortality (Figure 5A). Furthermore, the thrips feeding on R1 exhibited significant developmental retardation compared to those feeding on R2 or the control diet (Figure 5B).

### 3.4. Expression Levels of Fo-ILPs Are Correlated with Immature Development of F. occidentalis

The developmental retardation of the thrips feeding on the R1 hot pepper variety was analyzed by quantifying their *ILP* gene expression levels (Figure 6A). In comparison to R2 and the control diet, thrips feeding on R1 exhibited significantly lower (*p* < 0.05) expression levels of *Fo-ILP1* and *Fo-ILP2* during the larval and pupal stages. Interestingly, the thrips fed on the soybean diet showed rapid development and the highest expression levels of the two *ILP* genes. To determine the relationship between *ILP* expression and thrips development, the expression levels at the non-feeding pupal stage were compared with the developmental rates (Figure 6B). In both *ILP* genes, there were highly positive correlations between immature development and *ILP* expression.

## 4. Discussion

This study is a pioneering work on *ILP* genes and their physiological functions in Thysanoptera using *F. occidentalis*. In addition to a previous study on the *InR* of *F. occidentalis* [12], the identification of *ILP*s suggests that insulin signaling plays a crucial role in mediating the physiological processes of thrips. Specifically, the current study tested the roles of two *ILP*s in mediating the immature growth of *F. occidentalis*.

The two *ILP*s were expressed at stable levels at all developmental stages, from larvae to adults, except at the non-feeding prepupal and pupal stages. In investigating the low expression levels in the non-feeding stages, starvation treatment was applied to the larvae and adults, which led to suppression of the gene expressions. These observations regarding the reduction of the expression levels suggest that the nutrient signals derived from the feed diet induce expression of the *ILP* genes in *F. occidentalis*. Although the source of the *ILP*s was not investigated in *F. occidentalis*, it has been well documented that insect *ILP*s are produced in the brain, specifically at the insulin-producing cells (IPCs) in the pars intercerebralis [19]. The feed diet and subsequent digestion by the insect gut release nutrients into the hemolymph. These nutrients are sensed by the fat body, which sends a nutritional signal called the fat-body-derived signal to the brain in *Drosophila* [7]. Indeed, an insect leptin called Unpaired 2 is released from the fat body in response to feeding and activates *ILP* secretion from the IPCs through GABAergic neurons [20]. Furthermore, the fat body can directly sense glucose and release a nutrient signal called CCHamide-2 to activate the IPCs to secrete *ILP*s [21]. In *F. occidentalis*, the secreted *ILP*s may bind to the transmembrane protein *InR* to trigger intracellular insulin signals, which are presumably conserved in metazoans [22].

The suppression of the *ILP* gene expressions led to developmental retardation in *F. occidentalis* even under *ad libitum* feeding conditions. This suggested the roles of the two *ILP*s in mediating the immature development of *F. occidentalis*. The insulin signaling mediates various physiological processes, including immature growth and metabolism, in both invertebrates and vertebrates [5,23]. Additionally, the insulin signaling pathway is highly conserved in animals [22]. Further, once the *ILP*s bind to the *InR*, the downstream insulin signaling pathways are triggered by activating the insulin receptor substrates, which recruit phosphoinositide-3-kinase to the membranes to phosphorylate phosphatidylinositol-4,5-bisphosphate into phosphatidylinositol-3,4,5-trisphosphate (PIP3). The accumulated PIP3 levels then recruit phosphoinositide-dependent kinase, which activates serine-threonine protein kinase (Akt). Akt then phosphorylates Forkhead Box O (FOXO) to prevent its translocation into the nucleus while it phosphorylates the tuberous sclerosis complex (TSC) to upregulate protein translation by releasing the eukaryotic initiation factor 4E from its inhibitor 4E-binding protein. The amino-acid-mediated target of rapamycin (TOR) pathway can activate the TSC and share its downstream signal with the insulin signal. Thus, the induction of *ILP* expression in *F. occidentalis* would trigger insulin signaling in the target cells by the functional binding of *ILP* to *InR*, which activates a sequence of highly conserved downstream proteins to facilitate cell growth [19,24]. Our previous study on insulin signaling showed that *InR* expression is negatively correlated with the developmental rate of *F. occidentalis* [12]. It is well documented that *InR* expression is associated with immature development and adult reproduction in insects of the lepidopteran [25,26,27] and hemipteran species [28]. As predicted in the induction of the *InR* expression under starvation, the transcriptional factor FOXO could be phosphorylated and localized in the nucleus to survive under the harsh conditions [29]. Once insulin signaling is resumed, Akt phosphorylates FOXO to maintain the protein in the cytosol and activates the protein translation initiation factor for cell growth [30], which would lead to the immature development of *F. occidentalis*. This is further supported by the co-clustering of the two *ILP* genes with *Dm-ILP1*, which is known to mediate fly growth [31].

The developmental rates of *F. occidentalis* vary depending on its host plant varieties. The susceptibility of the host plants to thrips appeared to not be directly associated with resistance to TSWV infection. The insect resistance of the hot pepper to thrips was not clearly understood at the molecular level. However, it may be explained by inhibition of the thrips feeding behaviors because the varieties resistant to thrips exhibited much less feeding damage compared to the susceptible varieties. The hot peppers are resistant to another sucking insect, *Myzus persicae*, and deposit callose in the sieve elements of the phloem to prevent insect feeding behaviors [32]. It was an interesting observation that the expression levels of the *ILP* genes fluctuated with different host variety treatments. The plant variety facilitating larval development in *F. occidentalis* showed high *ILP* expression levels, whereas the variety retarding thrips development suppressed *ILP* expressions in *F. occidentalis*. These findings support the positive correlation between *ILP* expression and immature growth rate. The nutritional values of the different hot pepper varieties of *F. occidentalis* remain unknown. However, the severity of damage due to thrips feeding was highly associated with the preferences of the thrips for the host plants. Thus, the variety susceptible to thrips must have been consumed more than the resistant variety. The increased feeding amounts may be interpreted using the elevated *ILP* expression levels, which promote immature development through insulin signaling. However, further experiments would be necessary to elucidate the cause for differential *ILP* expression by assessing the composition of different nutrients as well as secondary metabolites in the different hot pepper varieties.

The positive correlation between *ILP* expression and immature growth suggests a regression to predict the growth rate of *F. occidentalis*. Our previous study on the negative correlation between *InR* expression and growth rate provides a regression formula: Developmental rate (1/day) = −0.035 × (*InR* expression) + 0.26. In *ILP* expression, *ILP1* shows a higher regression coefficient than *ILP2* in *F. occidentalis*, whose regression formula is as follows: Developmental rate (1/day) = 0.335 × (*ILP1* expression) + 0.102. Owing to the direct physiological interactions between insulin signaling and growth rate, several mathematical models have been developed to explain the functional interactions [33]. In addition, a comprehensive mathematical model of insect growth rate has been devised as a qualitative model using the insulin signaling pathway components [34]. The two components *ILP* and *InR* can thus be used to establish a novel insect growth model for *F. occidentalis* using the functional parameters. In considering multiple *ILP*s in insects, additional *ILP*s were predicted in *F. occidentalis*. For example, an *ILP* dissimilar to *Fo-ILP1* and *Fo-ILP2* was reported in the adult stage and inducible by ovarian development [35]. This suggests additional *ILP*s may be involved in the growth and development of *F. occidentalis*.

## Figures and Tables

**Figure 1 insects-14-00047-f001:**
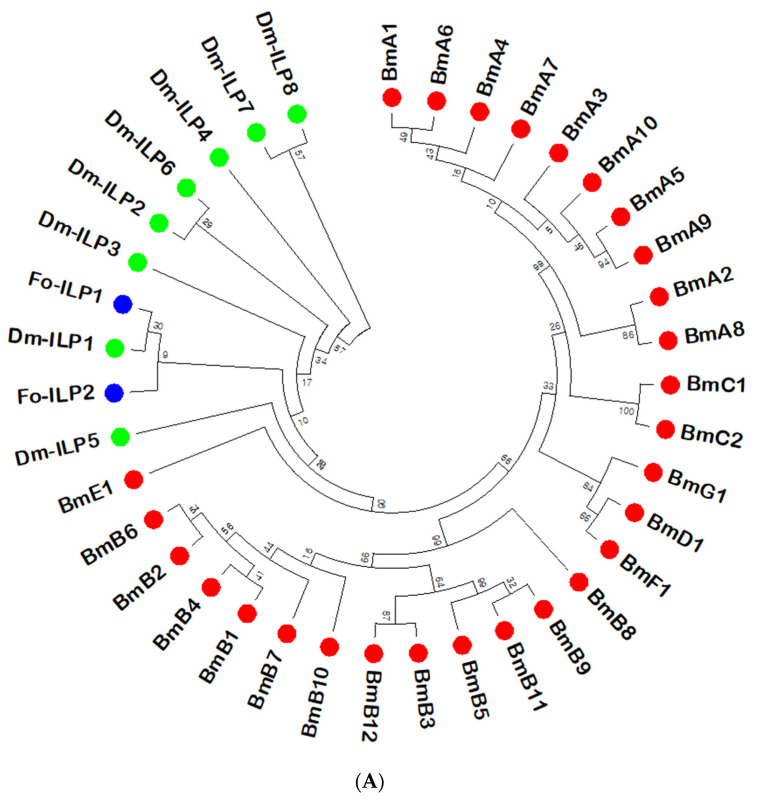
Two insulin-like peptides (*ILP*s; *Fo-ILP1* and *Fo-ILP2*) of *Frankliniella occidentalis*. (**A**) Phylogenetic analysis of the two *ILP*s with the eight *ILP*s of *Drosophila melanogaster* (DmILP1~Dm-ILP8) and 28 ILPs of *Bombyx mori* (BmA1~BmA10, BmB1~Bm12, BmC1, BmC2, BmD1, BmE1, BmF1, and BmG1). The tree was generated by the neighbor-joining method using MEGA 6.0. Bootstrapped values were obtained with 1000 repetitions to support branching and clustering. Amino acid sequences were retrieved from GenBank, and the accession numbers of the genes are shown in Appendix A. (**B**) Domain analyses of *Fo-ILP1* and *Fo-ILP2* compared to the known domains of *Dm-ILPs* [13]. ‘*’ indicates conserved cysteine residues at the B chain (marked with yellow squares) and the A chain (marked with green squares). Putative proteolytic cleavage sites are marked with blue squares. Signal peptides (marked with red squares) were predicted using the SignalP 5.0 server (https://services.healthtech.dtu.dk/service.SignalP-5.0/ (accessed on 31 December 2022)). Putative cleavage sites were predicted using the Neuroproteomics and Neurometabolomics Center on Cell–Cell Signaling (http://stagbeetle.animal.uiuc.edu/cgi-bin/neuropred.py (accessed on 31 December 2022)).

**Figure 2 insects-14-00047-f002:**
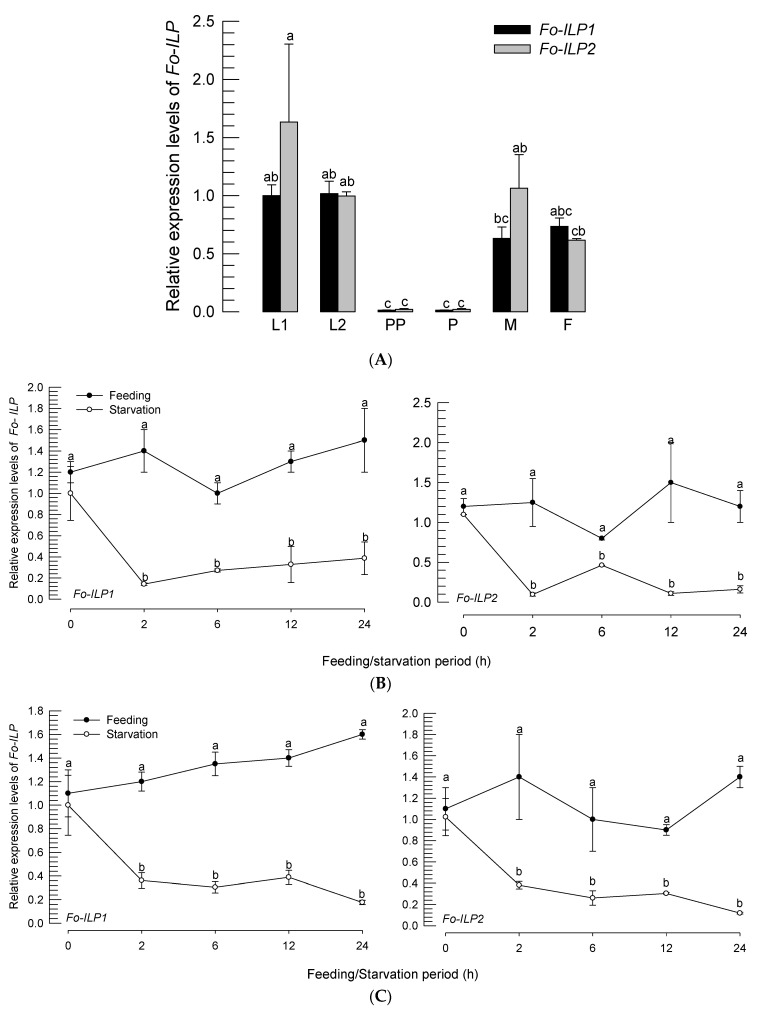
Expression profiles of the two *ILP*s (*Fo-ILP1* and *Fo-ILP2*) in *F. occidentalis*. (**A**) Expression patterns at different developmental stages: first and second instars (‘L1’ and ‘L2’), prepupa (‘PP’), pupa (‘P’), male adult (‘M’), and female adult (‘F’). (**B**) Effect of starvation on *ILP* expression in L2 larvae. (**C**) Effects of starvation on *ILP* expressions in female adults. Each treatment used 25 individuals for cDNA preparation and was independently replicated three times. The letters above the standard deviation bars in each gene indicate significant differences among the means at Type I error = 0.05 (LSD test).

**Figure 3 insects-14-00047-f003:**
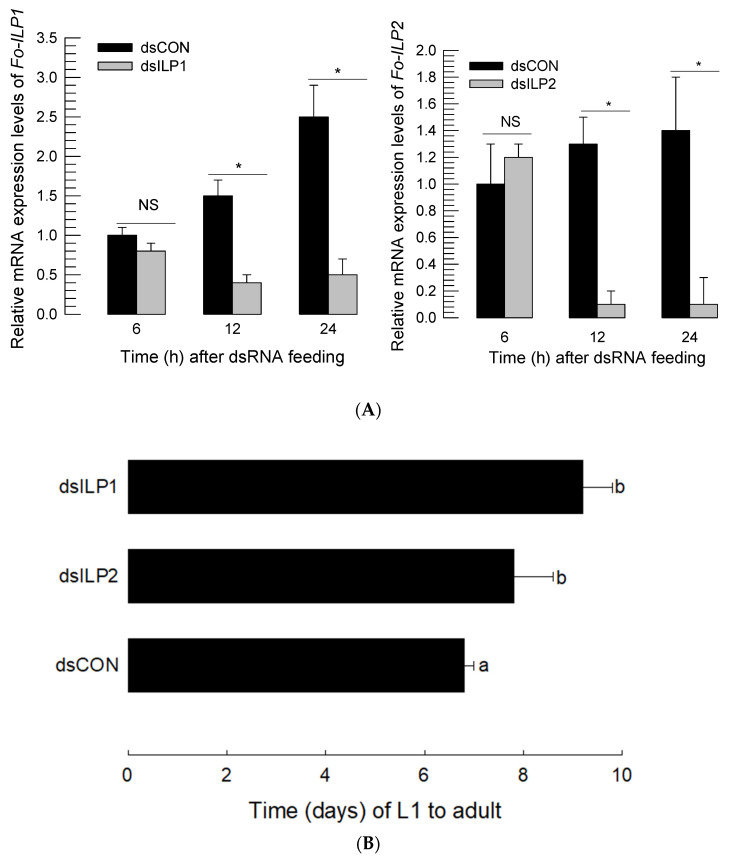
Effects of RNA interference (RNAi) of *ILP* expression on the immature development of *F. occidentalis*. RNAi was performed by feeding dsRNA specific to each *ILP* gene of *F. occidentalis*. (**A**) RNAi efficacy measured by mRNA levels using RT-qPCR. Each treatment used 25 individuals for cDNA preparation and was independently replicated three times. The asterisks above the standard deviation bars indicate significant differences between two treatments at type I error = 0.05 (LSD test). ‘NS’ stands for no significant difference. (**B**) Change in the immature developmental period from egg hatching (L1) to adult emergence by the RNAi treatments. ‘dsCON’ represents dsRNA specific to *EGFP*, which is a nontarget gene. Each treatment used 30 individuals to measure their developmental periods. The letters above the standard deviation bars indicate significant differences among the means at type I error = 0.05 (LSD test).

**Figure 4 insects-14-00047-f004:**
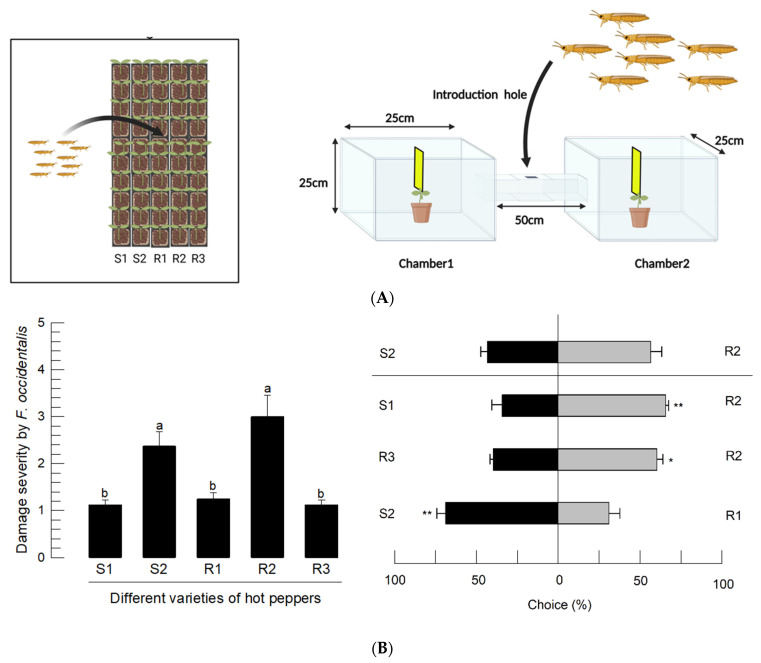
Variations among different hot pepper varieties for susceptibility to *F. occidentalis*. ‘S’ and ‘R’ stand for susceptibility or resistance to TSWV infection. (**A**) Feeding damage analysis of the hot pepper varieties to *F. occidentalis*. After 100 adults were released into the plastic bag (100 L), they were allowed to choose and feed on the hosts. After 7 days, feeding damage severity was graded using the criterion described in Appendix A. Each treatment was replicated three times. The letters above the standard deviation bars indicate significant differences among the means at type I error = 0.05 (LSD test). (**B**) Choice test of *F. occidentalis* adults between two hot pepper varieties via a two-armed tunnel assay. Different varieties of hot peppers were placed in each of the chambers with a yellow sticky trap to capture the thrips. A hundred adults were released at the entrance hole. Each treatment was replicated four times by exchanging the left and right positions. The asterisks above the standard deviation bars indicate significant differences between the means at Type I error = 0.05 (*) or 0.01 (**) (LSD test).

**Figure 5 insects-14-00047-f005:**
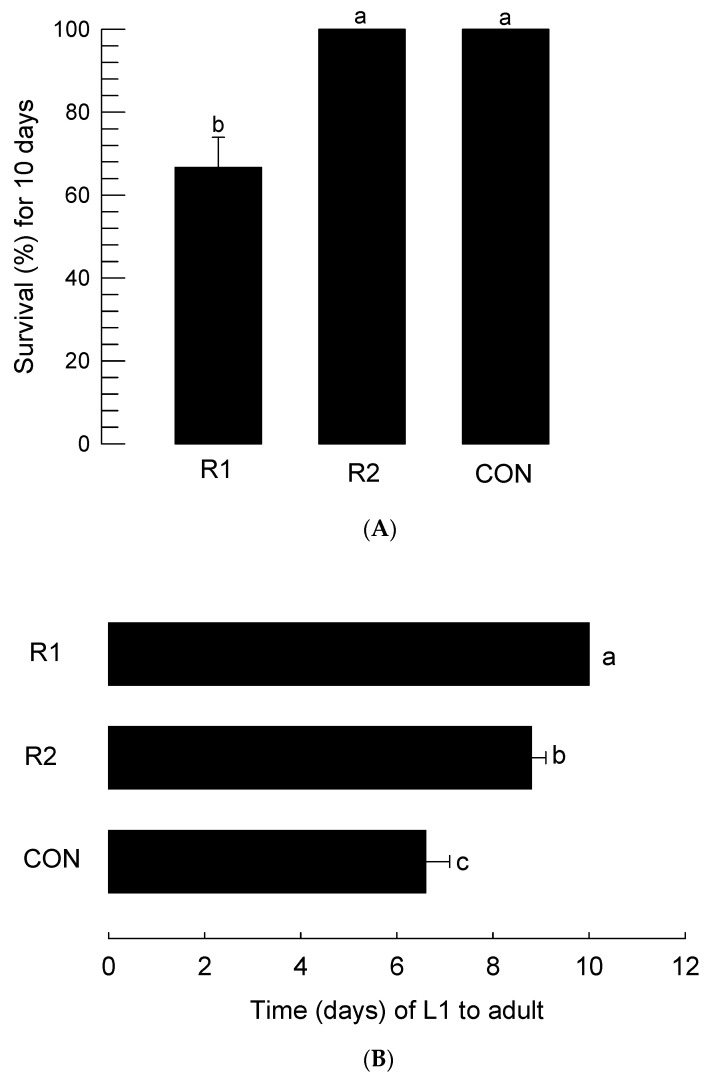
Influences of different varieties of hot peppers on the immature development of *F. occidentalis*. ‘R1’ and ‘R2’ are varieties resistant to TSWV infection. ‘CON’ represents the control based on the soybean diet. Leaf disks of the hot peppers were used for the feeding treatment of the thrips larvae. (**A**) Survival rate after feeding treatment for 10 days. Each treatment used 30 larvae and was replicated three times. (**B**) Changes in the immature developmental period from egg hatching (L1) to adult emergence by the feeding treatments. Each treatment used 20 individuals to measure their developmental periods. The letters above the standard deviation bars indicate significant differences among the means at type I error = 0.05 (LSD test).

**Figure 6 insects-14-00047-f006:**
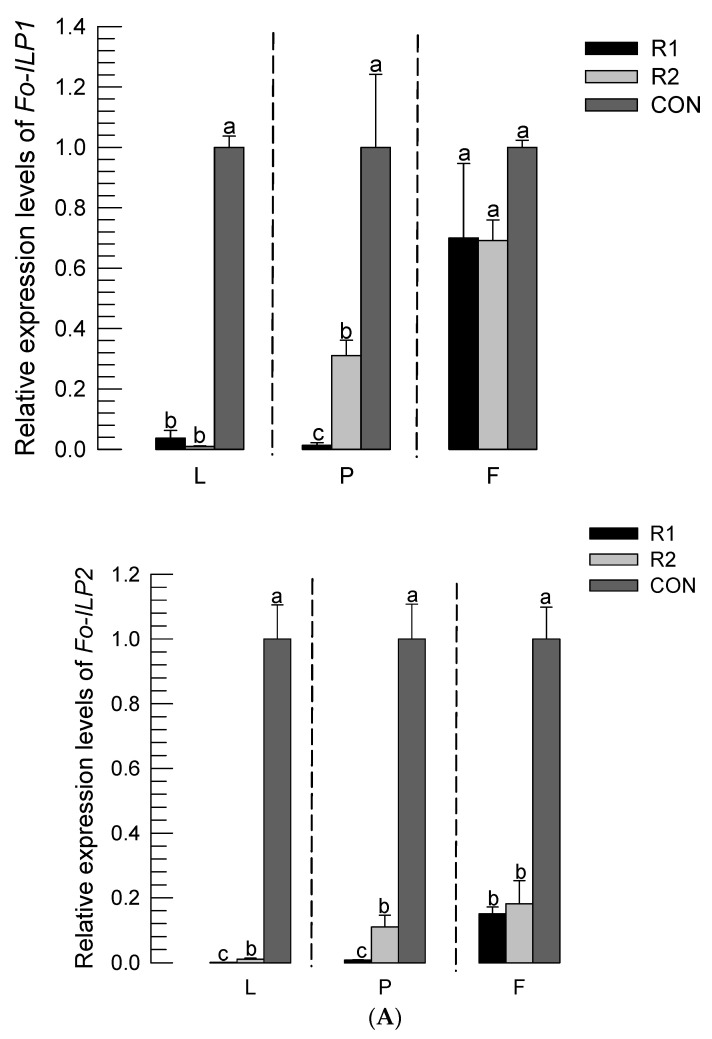
Influences of different varieties of hot peppers on the expressions of the two *ILP* genes of *F. occidentalis*. ‘R1’ and ‘R2’ are varieties resistant to TSWV infection. ‘CON’ represents control based on the soybean diet. Leaf disks of the two hot peppers were used for the feeding treatment of the thrips larvae. (**A**) Changes in the expression levels of the two *ILP*s (*Fo-ILP1* and *Fo-ILP2*) in *F. occidentalis*. RT-qPCR was performed on the first days of the second instar larvae (‘L’), pupae (‘P’), and adult females (‘F’). The letters above the standard deviation bars indicate significant differences among the means at type I error = 0.05 (LSD test). (**B**) Regression analysis between the developmental rate and *ILP* expression levels. The developmental rate was estimated by the inverse of the developmental period. *ILP* expression levels were obtained from the pupal stage. Each treatment used 20 larvae to measure the developmental periods.

## Data Availability

DNA and protein sequences are found in GenBank under the accession numbers given in the Methods section.

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
