# Peer review of "Insulin-like Peptides of the Western Flower Thrips Frankliniella occidentalis and Their Mediation of Immature Development"

_insects, 2023, doi:10.3390/insects14010047_

Round 1

Reviewer 1 Report

Dear Authors,

I have received for review:  Kim, C.-Y.; Kim, Y. Insulin-like peptides of the western flower thrips Frankliniella occidentalis and their mediation of immature development. I carefully read and studied the manuscript and my overall impression is that it is a well written article, containing valuable new information. The set up of experiments and their implementation is of high quality, with some exceptions as listed below. My opinion is that with some minor revision/correction it can be accepted for publication.

The references are correct and found no error or miss references, which is rather rare.

I will go by line numbers making my comment sor suggestions, offered replacements.

L46: replace at with in (the insulin-producing cells....)

L47: foods should be singular: food

L111: ...according to manufacturer’s instruction.

L117: instead of Table S3 Table S1 should be the first one. Move current Table S3 content to the first place (primers) 

L133: Table S3 is a wrong reference

L147: Table S1 content should be moved to S3 place and referred as Table S3

L168: write out in full Frankliniella

L180: using

L175: The GenBank references are in Table S2 (instead of S3). Morover Fo-ILP1 and Fo-ILP2 sequences are mixed accidently, as they are presented in an opposite way originally in GenBank database. Please check and correct accordingly.

L191: please remove „see”

L213: Why L1 were not checked or mentioned while on Fig. 2 results are shown? Please explain under 2.9 more clearly sample preparations.

L218: 2.10. Statistical analysis. Overall, in general LSD tests at a type-I error of 0.05 is mentioned, while later on figures (and legends) 0.01 and 0.001 also comes up using asterixs for indication. Using both asterixs (one, two or three) for different levels of significance, while using letters also are more than confusing. Please decide on one type of indication.

L252: Fig. 2A: Above the prepupa and pupa results a „c” represents that data are not significantly different, while the opposite is true, they are significantly different (Fig 2A). Please check statistical analysis as visibly there are significant differences. 

L253: patterns

L255: Effects

L262: Figure 3: Here 3 astexs are used for significant differences while p values are only 0.05. This is rather a one asterix if using LSD test. As mentioned above please use uniform indicators for statistical differences.

L303: Fig. 4B, lower: Standard deviation bars are missing.

L330: Fig. 5: Standard deviation bars are missing. However it is possible that that survival is 100%, however in this case ANOVA cannnot be used, since it can be employed where deviations are similar (homogeneity of variance), which is examined by Levene test. Did you perform normality tests? Under 2.10. Statistical analysis there is no mention of performing such test. Please check and correct!

L406: ILP expression levels were obtained from the pupal stage. Why only pupae were used for reference? What about the larval stages? Please delete last sentence under L406-407, it’s a repetiotion.

L417-419: Please replace as: The two ILPs were expressed at stable levels at all developmental stages from larva to adults, except at the non feeding prepupal and pupal stages. To investigate....

L436: What is meant under ad lib food supply in this case?

Currently I have no more remarks. Please carefully check the Supplementary material and the questions regarding significance and its interpretatation.

Kind regards,

A reviewer

Author Response

Comment #1-1: L46: replace at with in (the insulin-producing cells....)

Response: corrected as recommended

Comment #1-2: L47: foods should be singular: food

Response: corrected as recommended

Comment #1-3: L111: ...according to manufacturer’s instruction.

Response: corrected as recommended

Comment #1-4: L117: instead of Table S3 Table S1 should be the first one. Move current Table S3 content to the first place (primers) 

Response: corrected as recommended

Comment #1-5: L133: Table S3 is a wrong reference

Response: corrected as recommended

Comment #1-6: L147: Table S1 content should be moved to S3 place and referred as Table S3

Response: corrected as recommended

Comment #1-7: L168: write out in full Frankliniella

Response: corrected as recommended

Comment #1-8: L180: using

Response: corrected as recommended

Comment #1-9: L175: The GenBank references are in Table S2 (instead of S3). Morover Fo-ILP1 and Fo-ILP2 sequences are mixed accidently, as they are presented in an opposite way originally in GenBank database. Please check and correct accordingly.

Response: The table number is corrected. Fig. 1B is replaced with a new version.

Comment #1-10: L191: please remove „see”

Response: removed

Comment #1-11: L213: Why L1 were not checked or mentioned while on Fig. 2 results are shown? Please explain under 2.9 more clearly sample preparations.

Response: After feeding from L1 and the change in the expression levels were monitored in subsequent developmental stages. That is why we measured from L2.

Comment #1-12: L218: 2.10. Statistical analysis. Overall, in general LSD tests at a type-I error of 0.05 is mentioned, while later on figures (and legends) 0.01 and 0.001 also comes up using asterixs for indication. Using both asterixs (one, two or three) for different levels of significance, while using letters also are more than confusing. Please decide on one type of indication.

Response: corrected as follows: “Type I error of 0.05. Frequency data were analyzed by the mean difference test using PROC FREQ at Type I errors of 0.05 (*) and 0.01 (**).”

Comment #1-13: L252: Fig. 2A: Above the prepupa and pupa results a „c” represents that data are not significantly different, while the opposite is true, they are significantly different (Fig 2A). Please check statistical analysis as visibly there are significant differences. 

Response: To be clear, the statistical analysis is added as follows: “The letters above the standard deviation bars in each gene indicate significant differences among the means at Type I error = 0.05 (LSD test).”

Comment #1-14: L253: patterns

Response: corrected as recommended

Comment #1-15: L255: Effects

Response: corrected as recommended

Comment #1-16: L262: Figure 3: Here 3 astexs are used for significant differences while p values are only 0.05. This is rather a one asterix if using LSD test. As mentioned above please use uniform indicators for statistical differences.

Response: corrected as single asterisk

Comment #1-17: L303: Fig. 4B, lower: Standard deviation bars are missing.

L330: Fig. 5: Standard deviation bars are missing. However it is possible that that survival is 100%, however in this case ANOVA cannnot be used, since it can be employed where deviations are similar (homogeneity of variance), which is examined by Levene test. Did you perform normality tests? Under 2.10. Statistical analysis there is no mention of performing such test. Please check and correct!

Response: In Fig. 4B, SD is added. For the statistical analysis, we add the following statement in M&M: “The percentage data were arcsine-transformed to be normalized, and data obtained from the feeding damage or tunnel tests were subjected to a one-way analysis of variance using PROC GLM in SAS (SAS Institute, 1989).”

Comment #1-18: L406: ILP expression levels were obtained from the pupal stage. Why only pupae were used for reference? What about the larval stages? Please delete last sentence under L406-407, it’s a repetiotion.

Response: To make clear, the text is replaced with followings: “To get the relationship between ILp expression and thrips development, the expression levels at non-feeding pupal stage were compared with the developmental rates (Fig. 6B). In both ILP genes, there were highly positive correlations between immature development and ILP expression.”

Comment #1-19: L417-419: Please replace as: The two ILPs were expressed at stable levels at all developmental stages from larva to adults, except at the non feeding prepupal and pupal stages. To investigate....

Response: corrected as recommended

Comment #1-20: L436: What is meant under ad lib food supply in this case?

 Response: replaced as ad libitum 

Reviewer 2 Report

In this paper, the authors performed expression and functional analyses of ILPs in the western flower thrips. I think the data presented in this study is interesting, and the manuscript is clearly written.

This paper needs revision before publishing in this journal. My comments are listed below.

Major comments

Varieties of hot peppers.

This study showed that there is difference in ILP expressions among different hot pepper varieties. However, further experiments will be necessary to elucidate the cause for differential ILP expression. For example, composition of nutrients as well as secondary metabolites should be analyzed in these varieties. Without additional information, the data on hot pepper varieties remains incomplete. I would recommend the authors to do additional experiments so that this paper will be with higher quality.

Minor comments

Please cite the following reference.

Choi DY, Kim Y. 

Transcriptome analysis of female western flower thrips, Frankliniella occidentalis, exhibiting neo-panoistic ovarian development. 

PLoS One. 2022; 17(8):e0272399. 

Then, please clarify if either Fo-ILP1 or ILP2 in this study is tidentical to the one reported in the previous study.

L140, “test thrips were applied”

Please describe the developmental stage of the test thrips.

L175

“shown in Table S2”. Please revise.

L255-256

Please remove hyphen in “Effects” and “preparation”.

L253 and L418

I think the terms “propupa” and “propupal stage” are more frequently used rather than “prepupa” and “prepupal stage”.

Fig. 2 B and C

Please revise so that the horizontal axis is proportional to the duration (hours).

Table S2

The accession number of Fo-ILP1 is XP_026281410.1. 

Please revise.

Fig. S1.

Please point damage on leaves by arrows or circles.

Author Response

Comment #2-1: Varieties of hot peppers.

This study showed that there is difference in ILP expressions among different hot pepper varieties. However, further experiments will be necessary to elucidate the cause for differential ILP expression. For example, composition of nutrients as well as secondary metabolites should be analyzed in these varieties. Without additional information, the data on hot pepper varieties remains incomplete. I would recommend the authors to do additional experiments so that this paper will be with higher quality.

Response: This is a logical comment. We add this information to Discussion to make our study limitation: “However, further experiments would be necessary to elucidate the cause for differential ILP expression by assessing the composition of different nutrients as well as secondary metabolites in the different hot pepper varieties.”

Comment #2-2: Please cite the following reference.

Choi DY, Kim Y. Transcriptome analysis of female western flower thrips, Frankliniella occidentalis, exhibiting neo-panoistic ovarian development. PLoS One. 2022; 17(8):e0272399. Then, please clarify if either Fo-ILP1 or ILP2 in this study is tidentical to the one reported in the previous study.

Response: The reference is now added to Discussion as follows: “Considering multiple ILPs in insects, we predict additional ILPs in F. occidentalis. For ex-ample, an ILP dissimilar to Fo-ILP1 and Fo-ILP2 was reported in adult stage and inducible to ovary development (Choi and Kim, 2022). This suggests additional ILPs may be involved in growth and development of F. occidentalis.”

Comment #2-3: L140, “test thrips were applied” Please describe the developmental stage of the test thrips.

Response: specified as follows: “……to which 25 test thrips at L1 larval stage were applied and allowed to feed for 12 h.”

Comment #245: L175 “shown in Table S2”. Please revise.

Response: corrected as recommended

Comment #2-5: L255-256 Please remove hyphen in “Effects” and “preparation”.

Response: corrected as recommended

Comment #2-6: L253 and L418 I think the terms “propupa” and “propupal stage” are more frequently used rather than “prepupa” and “prepupal stage”.

Response: We do understand the terminology issue because the thrips undergo incomplete metamorphosis. However, a number of papers dealing with thrips used prepupa. In our previous papers, we used prepupa. Thus, we would like to use prepupa.

Comment #2-7: Fig. 2 B and C Please revise so that the horizontal axis is proportional to the duration (hours).

Response: corrected as recommended

Comment #2-8: Table S2 The accession number of Fo-ILP1 is XP_026281410.1. Please revise.

Response: Figure 1B is revised to match the GenBank accession number.

Comment #2-9: Fig. S1. Please point damage on leaves by arrows or circles.

Response: Arrows are added. Also, the figure caption is revised as follows: “Thrips damage had bright colored leaves (see arrows) and was classified into 6 grades”